# The Biological Function of Sigma-2 Receptor/TMEM97 and Its Utility in PET Imaging Studies in Cancer

**DOI:** 10.3390/cancers12071877

**Published:** 2020-07-13

**Authors:** Chenbo Zeng, Aladdin Riad, Robert H. Mach

**Affiliations:** Department of Radiology, Perelman School of Medicine, University of Pennsylvania, Philadelphia, PA 19104, USA; zengc@pennmedicine.upenn.edu (C.Z.); Aladdin.Riad@pennmedicine.upenn.edu (A.R.)

**Keywords:** sigma-2 receptors, TMEM97, MAC30, cancer, cholesterol, PGRMC1, LDLR, [^18^F]ISO-1

## Abstract

The sigma-2 receptor was originally defined pharmacologically and recently identified as TMEM97. TMEM97 has been validated as a biomarker of proliferative status and the radioligand of TMEM97, [^18^F]ISO-1, has been developed and validated as a PET imaging biomarker of proliferative status of tumors and as a predictor of the cancer therapy response. [^18^F]ISO-1 PET imaging should be useful to guide treatment for cancer patients. TMEM97 is a membrane-bound protein and localizes in multiple subcellular organelles including endoplasmic reticulum and lysosomes. TMEM97 plays distinct roles in cancer. It is reported that TMEM97 is upregulated in some tumors but downregulated in other tumors and it is required for cell proliferation in certain tumor cells. TMEM97 plays important roles in cholesterol homeostasis. TMEM97 expression is regulated by cholesterol-regulating signals such as sterol depletion and SREBP expression levels. TMEM97 regulates cholesterol trafficking processes such as low density lipoprotein (LDL) uptake by forming complexes with PGRMC1 and low density lipoprotein receptor (LDLR), as well as cholesterol transport out of lysosome by interacting with and regulating NPC1 protein. Understanding molecular functions of TMEM97 in proliferation and cholesterol metabolism will be important to develop strategies to diagnose and treat cancer and cholesterol disorders using a rich collection of TMEM97 radiotracers and ligands.

## 1. Introduction

Sigma receptors represent a class of proteins that were originally classified as a subtype of the opiate receptors [1]. Subsequent studies revealed that sigma receptor binding sites represented a distinct class of receptors that are located in the central nervous system as well as in a variety of tissues and organs [2]. Radioligand binding studies and biochemical analyses have shown that there are two types of sigma receptors [3,4]. Sigma-1 receptors (σ_1_Rs) have a molecular weight of 25 kD, whereas sigma-2 receptors (σ_2_Rs) have a molecular weight ranging from 18 to 21.5 kD. For many years, the sigma receptors were described pharmacologically through the binding of the radioligands [^3^H](+)-pentazocine and [^3^H]DTG (Figure 1). The radioligand [^3^H](+)-pentazocine has a high (3 nM) affinity for the σ_1_R and a low (>1000 nM) affinity for the σ_2_R, whereas [^3^H]DTG is equipotent at both σ_1_ and σ_2_ receptors. The σ_1_R was detected and measured by [^3^H](+)-pentazocine, whereas the σ_2_R by [^3^H]DTG in the presence of 100 nM unlabeled (+)-pentazocine to mask binding to σ_1_Rs. The σ_1_R was cloned in 1996 [5] and is a unique pharmacologically regulated integral membrane chaperone or scaffolding protein that allosterically modulates the activity of its associated proteins [6]. In contrast, the gene that codes for the σ_2_R remained unknown until recently.

## 2. Molecular Cloning of the σ_2_R

The σ_2_R was recently identified as transmembrane protein 97 (TMEM97, also known as MAC30) in 2017 [7]. Alon and colleagues synthesized a σ_2_R ligand and covalently coupled the ligand to agarose beads to prepare an affinity chromatography resin that was used to capture σ_2_Rs extracted from calf liver. Candidate proteins were isolated and identified by mass spectrometry. The candidate proteins were ectopically expressed and screened by σ_2_R binding assay using [^3^H]DTG. Among the candidate proteins tested, only TMEM97 resulted in a significant increase of specific [^3^H]DTG binding. Expression of TMEM97 in cells lacking σ_2_Rs increased σ_2_R binding and siRNA knockdown of TMEM97 proportionally reduced σ_2_R binding, confirming that TMEM97 is the σ_2_R. This work also demonstrated that Asp29 and Asp56 are essential for ligand recognition. Cloning the σ_2_R allows the use of modern biological methods to study its functions as a therapeutic and diagnostic target.

Previously, Xu et al. proposed that the binding site for the σ_2_R resided in a protein complex containing the progesterone receptor membrane component 1 (PGRMC1) [8]. Subsequent studies challenged the validity of these results and the recent identification of TMEM97 as the gene for the σ_2_R seemed to substantiate these reports [9,10]. However, Riad et al. recently reported that TMEM97 formed a complex with PGRMC1 and the low density lipoprotein receptor (LDLR) and this trimeric complex was responsible for the rapid internalization of the low density lipoprotein (LDL) in HeLa cells [11]. These data demonstrate that PGRMC1 associates with TMEM97 physically and corroborates our previous report that the σ_2_R represents a binding site in the PGRMC1 protein complex.

## 3. Evaluation of the σ_2_R PET Radiotracer for Assessing Proliferative Status and Predicting Drug Response

### 3.1. Evaluation of σ_2_R PET Radiotracer for Assessing the Proliferative Status and Growth Rate of Solid Tumors

#### 3.1.1. Evaluation of the σ_2_R as a Proliferation Biomarker

In a pioneer study, our lab demonstrated that the σ_2_R is a promising biomarker of the proliferative status in cell culture and mouse models with subcutaneous injection of mouse mammary adenocarcinoma line 66 cells, labeling with BrdU over time and assessing the proliferation status using flow cytometry compared to expression of σ_2_R using radioligand binding assays [12,13]. Previous studies had shown that line 66 cells displayed exponential growth kinetics for a period of 4 days, followed by a plateau phase that remained stable up to day 14 [14] (Figure 2A). In vitro studies also revealed that the exponentially growing 66 proliferating (66P) cells had 50% of the population in the G_1_ phase of the cell cycle and >90% of the population labeled with [^3^H]thymidine after 24-h labeling, indicating the cells were actively proliferating, whereas plateau phase 66 quiescent (66Q) cells have >97% of the population in G_1_ and 2% of the population labeled with [^3^H]thymidine after 24-h labeling indicating the cells were not proliferating as indicated by the lack of [^3^H]thymidine incorporation into newly synthesized DNA, which occurs in the S phase. Furthermore, subsequent studies revealed that 100% of the 66Q cells can be recruited into the P-cell compartment [15]. These data indicate that this cell line is an appropriate in vitro model system for studying σ_2_R expression in proliferating and quiescent cells.

In cell culture models, σ_2_R Scatchard studies revealed that 66P cells had a 10 fold increase the σ_2_Rs/cell than 66Q cells [13] (Figure 2B). The kinetics study of the expression of σ_2_Rs in 66 cells revealed a rapid increase in the σ_2_R density during the exponential growth phase that leveled off during the early plateau phase, followed by a decrease during the late plateau phase. Despite the kinetics of σ_2_Rs expression and cell growth kinetics being similar, the decreased density in the late plateau phase was delayed until day 10–12 after subculturing the Q-cells. Taken together, these results indicated a prolonged quiescent period is necessary to maximize σ_2_R loss from these 66 cells. In the mouse model of the 66 tumors, the ratio of the number of σ_2_R per P-cell to the number of σ_2_R per Q-cell was 10.6, consistent with the result obtained in the cell culture model [12]. These data indicate that the σ_2_R is a biomarker of the proliferative status of solid tumors and radioligands targeting the σ_2_R have the potential to noninvasively assess the proliferative status of tumors and the transition from proliferating to quiescent status using positron emission tomography (PET).

#### 3.1.2. Development of σ_2_R Radioligands

The Mach group has developed numerous σ_2_R radioligands for in vivo imaging of proliferation status and in vitro σ_2_R binding studies [18]. A σ_2_R selective PET radioligand, *N*-(4-(6,7-dimethoxy-3,-4-dihydroisoquinolin-2(1*H*)-yl)butyl)-2-(2-^18^F-fluoroethoxy)-5-methylbenzamide ([^18^F]ISO-1; Figure 1) has been validated as a PET imaging biomarker of the proliferation status in rodent models and human clinical studies (as described in the sections below). Another iodine-125 labeled σ_2_R radioligand [^125^I]RHM-4 (Figure 1) has been developed for in vitro binding studies. HeLa cells were used for these studies as they express TMEM97 and PGRMC1 and are able to be genetically edited efficiently via CRISPR/Cas9. In vitro σ_2_R binding studies using two different radioligands, [^125^I]RHM-4 and [^3^H]DTG, in HeLa cells demonstrated that knocking out TMEM97 resulted in a complete reduction in binding of [^125^I]RHM-4 to TMEM97 knockout (KO) HeLa cells, as well as significant but incomplete reduction of [^3^H]DTG binding to TMEM97 KO cells [11]. The results demonstrate that [^125^I]RHM-4 specifically binds to the TMEM97 protein, whereas [^3^H]DTG mainly binds to TMEM97 with high affinity (Kd = 14.8 nM), but also binds to other sites/proteins with relatively low affinity (Kd = 302.0 nM). Thus, [^125^I]RHM-4 is a more specific TMEM97 radioligand than [^3^H]DTG.

#### 3.1.3. Clinical Studies with [^18^F]ISO-1

Dehdashti et al. conducted the first human clinical study and established the feasibility of using [^18^F]ISO-1 to image solid tumors in lymphoma, breast cancer and head and neck cancer [19]. Thirty patients with primary breast cancer (*n* = 13), head and neck cancer (*n* = 10) and lymphoma (*n* = 7) underwent [^18^F]ISO-1 PET. Time–activity curves reached a plateau value after 20 min post-i.v. injection of the radiotracer. [^18^F]ISO-1 uptake values correlated with tumor Ki-67, a gold standard proliferation biomarker. In the entire group, the tumor maximum standardized uptake (SUVmax) value and tumor-to-muscle ratio correlated significantly with Ki-67 (τ = 0.27, *p* = 0.04 and τ = 0.38, *p* = 0.003, respectively). The presence of a significant correlation between [^18^F]ISO-1 and Ki-67 makes this agent promising for evaluation of the proliferative status of solid tumors. An interesting observation of [18F]ISO-1 was its high level of protein binding (free fraction 1%) and its high metabolic stability [19].

McDonald et al. conducted the clinical trial of [^18^F]ISO-1 in women with primary breast cancer to determine the correlation of [^18^F]ISO-1 PET uptake and Ki-67 expression [20]. Twenty-eight women with 29 primary invasive breast cancers were evaluated. The [^18^F]ISO-1 uptake quantitated by SUVmax at 55–60 min post-i.v. injection was correlated with the Ki-67 score determined by immunohistochemistry. Tumors stratified into the high Ki-67 (≥20%) group had SUVmax greater than the low Ki-67 (<20%) group (*p* = 0.02). SUVmax exhibited a positive correlation with Ki-67 across all breast cancer subtypes (ρ = 0.46, *p* = 0.01, *n* = 29; Figure 3). The data demonstrated that [^18^F]ISO-1 uptake in breast cancer modestly correlates with Ki-67 and can be used to provide an in vivo measure of tumor proliferative status.

### 3.2. Evaluation of [^18^F]ISO-1 for the Monitoring and Predicting Response to Therapy

[^18^F]ISO-1 was evaluated for assessing the proliferative status. Shoghi et al. conducted PET imaging studies with [^18^F]ISO-1 in xenografts of mouse mammary tumor 66 [21]. The [^18^F]ISO-1 uptake was correlated with the in vivo proliferative status determined by flow cytometric measures of BrdU-labeled tumor cells (Figure 2B,C). Data showed a strong linear correlation between the [^18^F]ISO-1 uptake and the proliferation status (i.e., ratio of proliferating cells to quiescent cells), indicating that [^18^F]ISO-1 can be used to image the proliferative status. In addition, [^18^F]ISO-1 was evaluated for assessing the tumor growth rate in a chemically-induced (N-methyl-N-nitrosourea (MNU)) model of rat mammary carcinoma. Each rat was imaged with [^18^F]ISO-1 over a 10-week period at 2-week intervals to assess the σ_2_R status. MRI was used to monitor the tumor volume. The data showed that there was a significant correlation (R = 0.68, *p* < 0.003) between [^18^F]ISO-1 uptake and changes in tumor volume between consecutive MR imaging sessions. These data suggest that a given value of [^18^F]ISO-1 uptake provides a predictive measure of an expected change in tumor volume (i.e., growth rate), which attests to the tumor’s aggressiveness.

[^18^F]ISO-1 was used to assess the response of MNU-induced tumors to bexarotene, a retinoid X receptor agonist and vorozole, an aromatase inhibitor therapeutic [21]. [^18^F]ISO-1 uptake determined by PET was correlated with tumor regression or the progression pattern. [^18^F]ISO-1 uptakes at week 2 in most tumors decreased upon treatment, which was generally in agreement with changes in tumor volume. The data suggest that [^18^F]ISO-1 uptake changes can be used to monitor the early drug response. The time-courses of [^18^F]ISO-1 uptake during 8 weeks of treatment and 2 weeks of drug withdrawal varied among different tumors, suggesting that different tumors have different sensibilities to a given therapy. This information can be used to guide drug selection and adequate dosing so that patients without sufficient response to the therapy can benefit from switching to alternative treatments and/or more aggressive dosage. In addition, the time-courses of [^18^F]ISO-1 and ^18^F-labeled 2-fluoro-2-deoxy-D-glucose ([^18^F]FDG) uptake in the same rat upon therapy were different in some tumors. This observation is consistent with the notion that [^18^F]ISO-1 and [^18^F]FDG, imaging proliferative status and glucose metabolism, respectively, can provide different information about molecular processes. Thus, [^18^F]ISO-1 is a new valuable predictor of the drug response.

Cyclin-dependent kinase 4/6 inhibitor (CDK4/6i) in combination with endocrine-therapy has emerged as an important regimen of care for estrogen receptor (ER)-positive metastatic breast cancer [22]. Despite the efficacy of this combined approach, not all patients benefit from the combination therapy. Biomarkers that can monitor changes in tumor proliferation early in the course of therapy can be used to identify patients most likely to benefit from this combinational therapy versus endocrine therapy alone. Recently, Elmi et al. assessed the ability of two PET-proliferation tracers, [^18^F]ISO-1 and [^18^F]FLT, for evaluating the response to palbociclib, a CDK4/6i and fulvestrant, an ER-antagonist [23]. PET imaging of MCF7 xenografts showed a significant decrease in [^18^F]FLT on day 3 and maintained low uptake on day 14 after treatment in all treatment groups (palbociclib, fulvestrant and the combination of both). The decrease in [^18^F]FLT uptake correlated with S phase depletion (Figure 4). On the other hand, the PET imaging showed no changes in [^18^F]ISO-1 uptake in all treated mice on day 3 and a significant decrease in [^18^F]ISO-1 uptake upon the combination therapy (but not single agents therapy) on day 14 (Figure 4). The decrease in [^18^F]ISO-1 uptake corresponded to G_0_ arrest. These results indicate that [^18^F]FLT can be used to monitor response early in the course of treatment and [^18^F]ISO-1 PET can be used as a tool for measuring transition of cells from proliferating status to quiescent status, a relatively later stage of proliferation inhibition. The data demonstrate that [^18^F]FLT and [^18^F]ISO-1 PET have independent and complementary mechanisms in evaluating tumor-proliferation. Longitudinal [^18^F]FLT and [^18^F]ISO-1 PET might serve as a clinically translatable approach for predicting and monitoring response to combinatorial CDK4/6i and endocrine therapy in patients with ER-positive breast cancer to guide personalized treatment in oncology.

## 4. Biological Characteristics and Functions of σ_2_R/TMEM97

### 4.1. Subcellular Localization of σ_2_R/TMEM97

TMEM97 exists in multiple subcellular locations. (1) TMEM97 localizes in cellular membranes. Before the σ_2_R was cloned, it was detected and measured by the σ_2_R binding assay in membrane preparations of various tissues and cell lines [4,24]. In fact, the σ_2_R was purified and identified as TMEM97 from calf liver membrane fractions [7]. Based on the amino acid sequence of TMEM97, this protein is predicted to have a four-pass transmembrane topology. Together, these data indicate that TMEM97 is a membrane-bound protein. (2) TMEM97 localizes in lipid rafts. Gebreselassie and Bowen demonstrated that the σ_2_R, detected using [^3^H]DTG and a lipid raft biomarker flotillin-2, detected by immunoblotting, colocalized in lipid raft fractions after equilibrium centrifugation of extracts of rat liver membranes in discontinuous sucrose density gradients [25]. The data indicate that the σ_2_R localizes in lipid rafts. (3) TMEM97 localizes in the endoplasmic reticulum (ER), lysosomes and plasma membranes. TMEM97 sequence analysis shows that the C-terminus contains the predicted ER-retention sequence “KRKKK” [7]. Transient expression of TMEM97-YFP in HeLa cells revealed a predominant localization of the protein to reticular ER-like membranes and the nuclear envelope, consistent with ER localization. On the other hand, in sterol-depleted cells, a prominent fraction of TMEM97-YFP localized to lysosome and the plasma membrane. Sterol depletion-induced TMEM97 translocation is believed to regulate lysosomal cholesterol export. A confocal microscopy study showed that σ_2_R fluorescent ligands were colocalized with ER, lysosome and plasma membrane trackers, consistent with the aforementioned findings [26]. Subcellular localization of TMEM97 should provide clues to study TMEM97 biological functions.

### 4.2. Biological Characterization and Roles of TMEM97 in Cancer Proliferation

#### 4.2.1. Differential Expression of TMEM97 in Normal and Cancer Tissues

TMEM97 expression in normal and cancer tissues has been studied using an σ_2_R binding assay and immunohistochemistry to measure protein levels and using northern-blot, quantitative RT-PCR, DNA microarray or RNAseq methods to determine mRNA levels. Using σ_2_R binding assays, Vilner et al. first demonstrated that the σ_2_R exists in a number of rodent and human tumor cell lines [24]. Numerous studies demonstrated that TMEM97 was ubiquitously expressed in a wide variety of normal human tissues and cancer. TMEM97 mRNA and/or proteins was significantly increased in breast, colon, gastric, esophageal, lung, ovarian cancer and prostate cancer [27,28,29,30,31,32,33,34,35,36], but decreased in meningiomas, pancreatic and renal cancer [27,37]. TMEM97 upregulation in certain tumors and downregulation in others suggests that this protein plays a distinct role in human malignancies.

#### 4.2.2. Potential Requirement of TMEM97 in Cancer Proliferation

It is reported that stable knockout of TMEM97 by CRISPR/Cas9 gene editing decreased cell proliferation in prostate cancer LNCaP cells [36]. It is also reported that transient knockdown of TMEM97 by siRNA decreased cell viability and migration/invasion in breast cancer, gastric cancer and glioma cell [38,39,40]. These data suggest that TMEM97 may be required for cell proliferation in certain tumor cell lines.

#### 4.2.3. Regulation of TMEM97 Expression in Cancer

The TMEM97 gene is located on chromosome band 17q11.2. The coding nucleotides of TMEM97 in 39 ovarian cancer samples were sequenced [41]. No coding sequence variations were found, suggesting that the TMEM97 coding sequence is not responsible for ovarian cancer development.

Limited research examined the regulation of TMEM97 expression in cancer. (1) TMEM97 was found to be a transcriptional target of the BRCA1 gene [42]. Inherited mutations of the BRCA1 gene predispose to breast, ovarian and other cancers. The role of the BRCA1 gene in the maintenance of chromosomal integrity is linked to transcriptional regulation function of the BRCA1 protein. In one study, differentially expressed genes were identified by comparing control MCF7 breast cancer cells with MCF7 cells ectopically expressing BRCA1. TMEM97 was among a set of BRCA1-induced genes. The data suggest that TMEM97 may be related to BRCA1 function in breast cancer cells. (2) The epidermal growth factor (EGF) transmitted signal transduction pathways play a major role in carcinogenesis and proliferation of human bladder cancer cells. One study demonstrated that treatment with 10 ng/mL EGF for 5 days downregulated TMEM97 mRNA expression in the human bladder cancer cell line J82, but had no effect on TMEM97 mRNA expression in another human bladder cancer cell line RT4 [43]. The data suggest that TMEM97 expression is differentially regulated by EGF in different bladder cancer cell lines. (3) TGF-β is known to be overexpressed in pancreatic cancer. The effects of TGF-β on TMEM97 mRNA expression were examined in pancreatic cancer cell lines [27]. TMEM97 was transiently downregulated by TGF-β in two pancreatic cancer cell lines Colo-357 and Panc-1. In contrast, TMEM97 expression was not affected by TGF-β in five other pancreatic cancer cell lines. The data suggest that TMEM97 expression is differentially regulated by TGF-β expression in different pancreatic cancer cell lines. (4) MicroRNAs are globally downregulated in prostate cancer, especially in poorly differentiated tumors. miR-152-3p was identified as a common epigenetically regulated onco-suppressor in prostate cancer [36]. miR-152-3p was under-expressed in prostate cancer. Overexpression of miR-152-3p suppressed cell viability and invasion potential and downregulated TMEM97 mRNA expression in prostate cancer LNCaP cells. Knockout of TMEM97 inhibited proliferation of LNCaP cells. The data indicate that TMEM97 is a target gene of miR-152-3p and may be related to prostate cancer development and proliferation. Taken together, TMEM97 is regulated by oncogenic factors in certain cancer cells and may be related to cancer development and proliferation in certain tumors.

#### 4.2.4. TMEM97 Does Not Mediate σ_2_R Ligand Cytotoxicity

σ_2_R ligands with various structures have been shown to induce cell death in cancer cells. Zeng et al. examined the role of TMEM97 and PGRMC1 in mediating σ_2_R ligand-induced cell death [44]. The data showed that knockout of TMEM97, PGRMC1 or both by CRISPR/Cas9 gene editing did not affect EC_50_ values, the concentrations of σ_2_R ligands by which 50% of cell death was induced (Table 1), suggesting that cytotoxic effects of these compounds are not mediated by TMEM97 and/or PGRMC1. It is of interest to note that not all σ_2_R ligands display cytotoxicity. For example, the σ_2_R ligands siramesine, SW43 and PB28 display similar cytotoxicity in the above described cell lines, whereas the benzamide analogs RHM-4 and ISO-1 are not cytotoxic under any conditions (Table 1) [42]. This work will facilitate elucidating mechanisms of σ_2_R ligand cytotoxicity.

### 4.3. Biological Roles of TMEM97 in Cholesterol Homeostasis

Several lines of evidence indicate that TMEM97 plays important roles in cholesterol homeostasis. Ovarian cancer most often derives from ovarian surface epithelial cells. Increased exposure to progesterone (P4) protects women against developing ovarian cancer. Wilcox et al. determined P4-induced gene expression in ovarian surface epithelial cells with oligonucleotide microarrays [41]. The data showed that TMEM97 and cholesterol biosynthesis genes were coordinately upregulated upon P4 treatment, suggesting that TMEM97 associates with cholesterol and lipid metabolism. The P4-induced alterations in cholesterol and lipid metabolism in ovarian surface epithelial cells might play a role in conferring protection against ovarian cancer.

Elevated plasma cholesterol levels result in excess cholesterol deposition in arterial vessel walls and are a major risk factor for atherosclerosis. In order to understand cellular cholesterol regulation, Bartz identified cholesterol-regulating genes by cultivating HeLa cells in sterol-depleted serum to activate cellular sterol homeostatic regulatory machinery and then performing genome-wide gene expression analysis with cDNA microarrays and targeted RNAi screening [45]. One of 20 genes identified with this strategy is TMEM97. The data showed that TMEM97 mRNA was regulated by upstream cholesterol regulatory factors: sterol depletion upregulated TMEM97 mRNA; low density lipoprotein receptor (LDLR) knockdown by siRNAs upregulated TMEM97 mRNA under control condition (in the presence of sterol) and sterol regulatory-element binding protein (SREBP-2) knockdown by siRNAs downregulated TMEM97 mRNA in sterol-depleted cells. The data also demonstrated that TMEM97 regulated downstream cholesterol homeostasis: knockdown of TMEM97 by siRNAs considerably inhibited cellular low density lipoprotein (LDL) uptake under sterol depletion condition and total free cholesterol under control condition. TMEM97 interacted with LDL cholesterol transport-regulating protein Niemann-Pick C1 (NPC1). NPC1 knockdown induced a robust cholesterol storage, whereas TMEM97 knockdown increased NPC1 protein level, reduced lysosomal lipid storage and restored cholesterol trafficking to the endoplasmic reticulum in NPC1-knockdown cells [46]. Thus, TMEM97 was identified as the cholesterol-regulating gene.

Riad et al. reported that TMEM97 and/or PGRMC1 knockout in HeLa cells reduced LDL internalization [11]. Immunocytochemistry and proximity ligation assay studies indicated that TMEM97 interacted with PGRMC1 and LDLR physically, suggesting that this ternary complex was responsible for this rapid rate of internalization of LDL.

By a computational protein sequence analysis, Sanchez-Pulido and Ponting identified a new domain, the expanded EBP superfamily (EXPERA) domain, which is conserved among TMEM97, transmembrane 6 superfamily member (TM6SF1 and TM6SF2), emopamil binding protein (EBP) and emopamil binding protein-like protein (EBPL) families [47]. This EXPERA domain family of proteins is implicated in cholesterol homeostasis. Carriers of the Glu167Lys coding variant in the TM6SF2 gene are more susceptible to non-alcoholic fatty liver disease (NAFLD). EBP is an enzyme with ∆8–∆7 sterol isomerase activity and regulates key steps in the final cholesterol biosynthesis pathway. EBP mutations are the cause of chondrodysplasia punctata 2 X-linked dominant (CDPX2), a defective cholesterol biosynthesis disorder. These evidences further indicate that TMEM97, like other EXPERA domain-containing proteins, is involved in cholesterol homeostasis.

### 4.4. TMEM97 in the Interplay of Cancer and Cholesterol Homeostasis

The involvement of TMEM97 in cholesterol homeostasis is consistent with the findings that TMEM97 is upregulated in proliferating versus quiescent breast cancer cells and TMEM97 is differentially expressed in normal and cancer tissues. Cholesterol is a key component of the cell membrane. Accelerated cholesterol and lipid metabolism are the hallmarks of cancer [48]. The metabolic dependency of cancer cells on cholesterol and other lipids is tightly regulated by the cholesterol homeostasis network, including (1) SREBP, master transcriptional regulators of cholesterol and fatty acid pathway genes, (2) nuclear sterol receptors (liver X receptors, LXR), which regulate intracellular cholesterol level through the expression of cholesterol efflux proteins such as ABC transporters and (3) lipid particle receptors, such as LDLR, providing exogenous sterol and lipids to cancer cells. Besides the well-established structural role of cholesterol in membranes, the deregulation of metabolite production at different points in the cholesterol bio-synthesis pathway has been reported to promote cancer or cause resistance to therapies in different in vitro and in vivo models described in [49]. TMEM97 is a target gene of SREBP and mediates LDL uptake, cholesterol trafficking and cellular cholesterol level. Given the metabolic dependency of cancer cells on cholesterol and various effects of cholesterol metabolites on cancer, TMEM97 may play roles in cancer via involvement in cholesterol homeostasis (Figure 5).

## 5. Conclusions

TMEM97 has been validated as a biomarker of proliferative status in in vitro and in vivo studies. [^18^F]ISO-1, which targets TMEM97, has been validated as a PET imaging biomarker of proliferative status in human breast cancer patients. [^18^F]ISO-1 is also a promising predictor of the drug response in preclinical studies. The proliferative status of a tumor has a profound effect on the outcome of both chemotherapy and radiotherapy treatments. Thus, [^18^F]ISO-1 should be useful to guide treatment plan for cancer patients.

TMEM97 plays distinct roles in cancer. TMEM97 is upregulated in some tumors but downregulated in other tumors. TMEM97 may be required for cell proliferation in certain tumor cells. TMEM97 is regulated by upstream oncogenic signals such as EGF, TGF-β and microRNA in some but not other tumor cells. The data suggest that TMEM97 is associated with cancer proliferation. Emerging evidence also demonstrates that TMEM97 plays important roles in cholesterol homeostasis. TMEM97 expression is regulated by upstream cholesterol-regulating factors such as sterol depletion and SREBP expression levels. TMEM97 regulates the downstream processes such as LDL uptake by forming complexes with PGRMC1 and LDLR, as well as cholesterol transport out of lysosome by interacting and regulating the NPC1 protein. Moreover, TMEM97 belongs to the EXPERA domain-containing protein family, which is implicated in cholesterol biology. Understanding molecular functions of TMEM97 in proliferation and cholesterol metabolism will be important to elucidate the interplay between cancer proliferation and cholesterol metabolism and to develop strategies to diagnose and treat cancer and cholesterol disorders using a rich collection of existing σ_2_R/TMEM97 radiotracers and ligands [18,50,51].

## Figures and Tables

**Figure 1 cancers-12-01877-f001:**
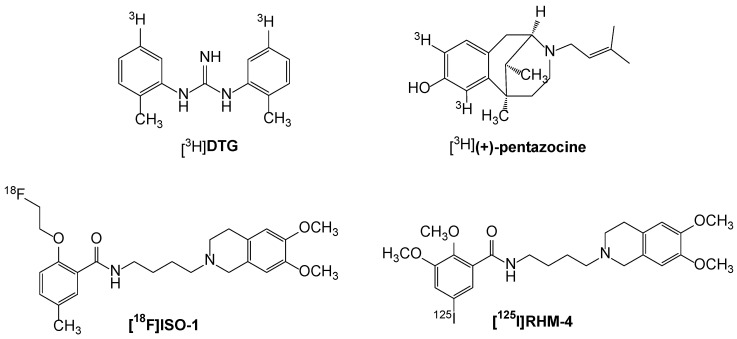
Structures of radioligands used in in vitro binding and PET imaging studies of the s2 receptor/TMEM97 in cancer patients.

**Figure 2 cancers-12-01877-f002:**
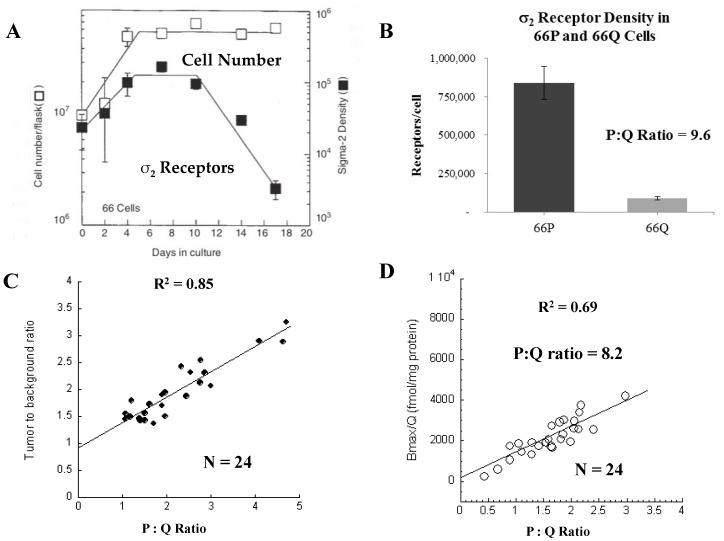
Relationship of the s2R with cell proliferation. (**A**) Comparison of the growth curve of 66 cells and the density of s2R in proliferating and quiescent cells. (**B**) Density of s2R in 66P versus 66Q cells. (**C**,**D**) Comparison of the tumor: Background ratio of [^18^F]ISO-1 uptake (**C**) and s2R density (**D**) versus the P:Q ratio of solid 66 tumors. Adapted from [16,17].

**Figure 3 cancers-12-01877-f003:**
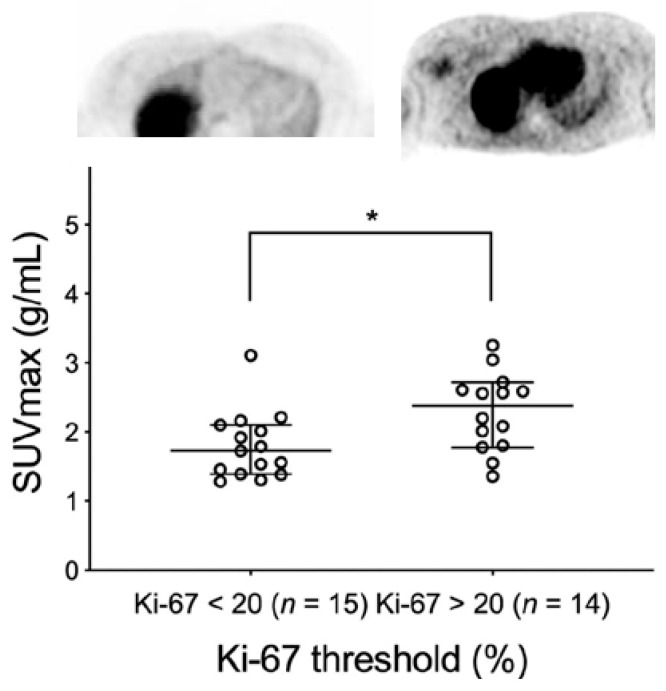
Uptake of [^18^F]ISO-1 in breast cancer patients. There was a higher uptake of [^18^F]ISO-1 in patients having a Ki-67 score >20. The image on the left is from a patient having a low Ki-67 (Ki-67 = 7.5%) score and low uptake of [18F]ISO-1, where the image on the right is from a patient having a high Ki-67 score (Ki-67 = 74%) and high uptake of [18F]ISO-1. The graph shows the significant difference in [18F]ISO-1 uptake in patients above and below the Ki-67 score of 20 (* *p* < 0.05). Data from reference [20].

**Figure 4 cancers-12-01877-f004:**
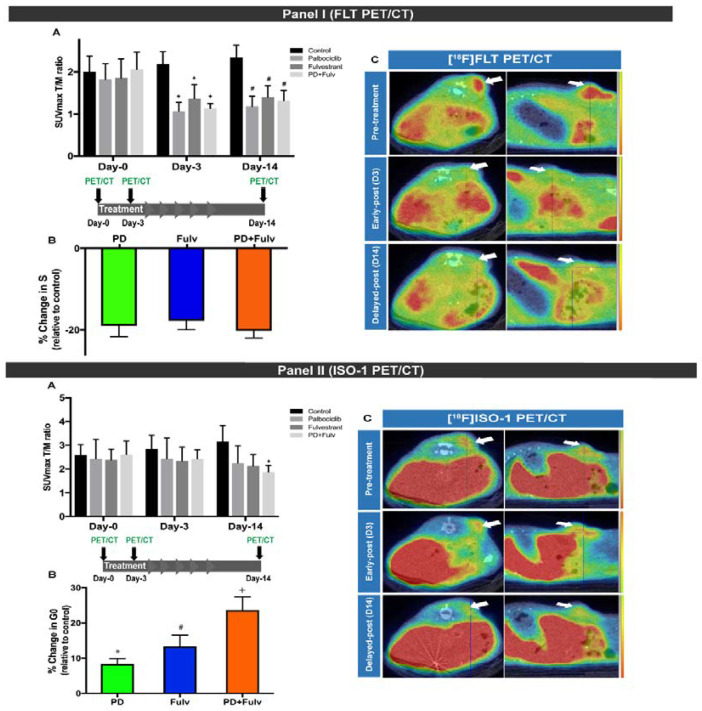
Panel (I): [^18^F]FLT uptake after treatment with palbociclib (PD), fulvestrant (Fulv) or combination of both at baseline and days 3 and 14 of treatment. (**A**) Tumor uptake as T/M ratio in controls and treated mice, showing persistent reduction of the T/M ratio in treated mice on days 3 and 14. (**B**) Changes in S-phase fraction relative to controls on day 14 of treatment, demonstrating comparable S-phase reduction across different treatment groups. (**C**) Representative PET/CT images of one mouse from the combination treatment group, demonstrating dramatic decline of [^18^F]FLT uptake on day 3, which is persistent on day 14. Panel (II): [^18^F]ISO-1 uptake after treatment with palbociclib (PD), fulvestrant (Fulv) or combination of both at baseline and days 3 and 14 of treatment. (**A**) Tumor uptake as the T/M ratio in controls and treated mice, showing the delayed reduction of the T/M ratio in treated mice, seen only on day 14. (**B**) G_0_ arrest relative to controls on day 14 of treatment, more pronounced in the combination therapy group. (**C**) Representative PET/CT images of one mouse from the combination treatment group, demonstrating delayed decline in [^18^F]ISO-1 uptake on day 14. *, *p* < 0.05; #, *p* < 0.01 and +, *p* < 0.001, compared with control group of the respective day. Reproduced with permission from reference [21].

**Figure 5 cancers-12-01877-f005:**
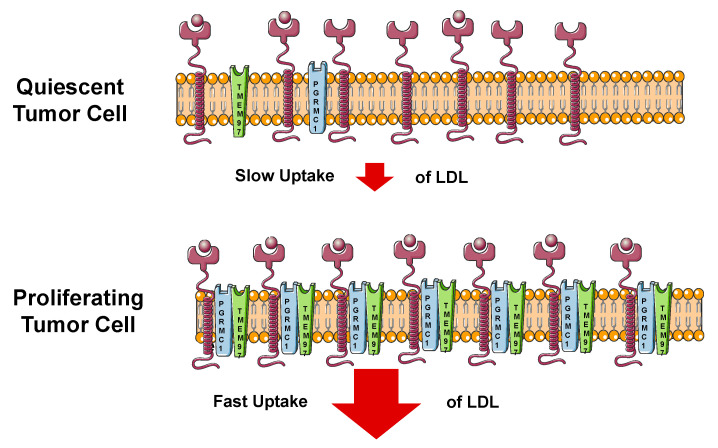
Cartoon showing the mechanism of uptake of the low density lipoprotein (LDL) in quiescent (slow) and proliferating (fast) tumor cells.

**Table 1 cancers-12-01877-t001:** EC_50_ values of s2R ligands in control, TMEM97 KO, PGRMC1 KO and double KO cell lines.

Ligand	Control (µM) Mean ± SD	TMEM97 KO (µM) Mean ± SD	PGRMC1 KO (µM) Mean ± SD	Double KO (µM) Mean ± SD
**Siramesine**	12.1 ± 1.7	10.3 ± 1.49	10.7 ± 0.7	10.0 ± 1.3
**SW43**	33.7 ± 8.6	30.4 ± 3.4	33.2 ± 1.6	32.8 ± 3.8
**PB28**	58.2 ± 5.9	59.8 ± 10.7	59.4 ± 8.0	57.4 ± 6.9
**RHM-4**	>200	>200	>200	>200
**ISO-1**	>200	>200	>200	>200

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
