# Peer review of "The Biological Function of Sigma-2 Receptor/TMEM97 and Its Utility in PET Imaging Studies in Cancer"

_cancers, 2020, doi:10.3390/cancers12071877_

Round 1

Reviewer 1 Report

The authors present a very interesting review on the imaging of the sigma-2 receptor/TMEM97 and is potential for use in imaging solid tumors. At times the manuscripts seems to be a review of the target and in others a review of the use of [18F]ISO-1 for imaging the target. This is natural as the interest is in using the imaging agent for monitoring expression of the target, but perhaps it should be reflected in that way in the title, the abstract and throughout?  When the concept of thymidine was first introduced I was confused based on the presentation if it was being proposed as a TMEM97 imaging agent. In section 3.1.1. make it clear how your lab explored for new markers of proliferation (perhaps start with the experiment design to make it clear to the reader). At the end of section 3.1.2 one uninitiated in imaging agents might wonder why RHM-4 was not further pursued. Provide a sentence to transition to the following section and why and how ISO-1 was developed, presumably you desired a PET imaging radioligand for better sensitivity and resolution; thus you developed one based on your best ligand RHM-4. A few statements on why the method of labeling (Fluoroethyl prosthetic) was selected and the metabolic stability, its biodistribution and radiation dosimetry are of interest to this reviewer and likely other readers. The authors describe how the target is involved in cholesterol trafficking, but do not provide a discussion on the suitability of models reported for studies involving a target where cholesterol biology is in question (e.g. for background see ref. J Biomed Res. 2015 Aug 20;30(1):3-10. doi: 10.7555/JBR.30.2015005 https://pubmed.ncbi.nlm.nih.gov/26585560/  ; Arterioscler Thromb Vasc Biol. 2012 May; 32(5): 1104–1115 PMID: 22383700). In general for each study, a statement on the rigor of the research would be good. Overall, I think this is a very good review and recommend it for publication once the comments are addressed.

Pg 2, Ln 66: As Mach is an author on this publication, I am not sure third person is warranted as it implies objectivity, which cannot be present. The authors should just refer to it as, "In our work..." or something related

Pg 2, Ln 68: The authors refer to a prior study using mouse adenocarcinoma cells but have indicated that the target of interest has to do with cholesterol trafficking in the body. It is well known that mice do not handle cholesterol in the same manner as humans, and that most studies regarding cholesterol pathways require rabbit models to show pharmcological action that is meaningful for human disease. This is due largely to that in mouse HDL is the major lipoprotein and have high hepatic LDLR expression; whereas, in humans and rabbits LDL is the major lipoprotein and less LDLR is present. Mice are also resistant to forming cholesterol disease from cholesterol diet and require a knockout of ApoE; in addition, mice also lack CETP which is important in cholesterol trafficking. Based on these differences the authors should provide justification beyond observed uptake of thymidine for this model in their study and state its strengths and weaknesses when it comes to predicting utility and function in human disease.

Pg 3, Ln 94: The authors title the section "development of σ2R radioligands", but then use the term radioligand and radiotracer interchangeably. Based on the data reported, the term radioligand is more appropriate despite the colloquial use of "radiotracer" that has occurred in the literature to describe all PET imaging agents (even though many are radioligands and some are even radioindicators). A radiotracer is typically defined as an agent that by virtue of its radioactive decay can be used to explore the mechanism of chemical reactions by tracing the path that the radioisotope follows from reactants to products (FDG being the classic example). While thymidine would qualify as a radiotracer (incorporation into DNA by chemical reaction) when radiolabeled the described ligands would not. Please make this clear in the manuscript.

Pg 4, Ln 127: An SUVmax level is given, please describe the experiment and the time of SUV max or any dynamic data/PK that was collected (was it early and corresponding to blood flow or would the data allow for more precise quantification). While in oncology PK is often not looked at during development to know if an agent is flow limited or amenable for quantification of binding: it would be interesting to know the thoroughness of the study. A minor point for this manuscript, but as this was a human study how were the mass of the human subjects taken into account given the known issues with using SUV as a measure in overweight to obese subjects?

Pg 4: Can the authors provide the biodistribution and the radiation dosimetry of [18F]ISO-1?

Pg 9: When mentioning the study please make it clear for each reference what the model of study was being reported: cellular assay in a well, xenograft on a mouse, spontaneous model in a mouse,  a different animal model (e.g. knockout studies are done in rabbits, as well as mice among other species), resected tumor from a human patient?

Pg 9, Ln 312-328: Given the target's importance in cholesterol trafficking are there any studies of its expression in adrenocortical cells of the adrenal gland (even biodistribution would be of interest) or ovaries given the high levels of cholesterol storage and steroid synthesis that occurs in those tissues. A specific mention of anything about its role in adrenocortical carcinoma would also be of interest.

Reviewer 2 Report

The presented manuscript is a review of "the biological function of sigma-2 receptor/TMEM97 2 and its utility in PET imaging studies in cancer".  The authors did a nice job of collecting information. Unfortunately, the manuscript in a current form is barely ready for publication. The flow of the manuscript is erratic and difficult to follow. Some of the terms appeared without explanation. The terminology through the text is changing. Also, there are several misleading or wrong references, direct copy-paste from textbooks, and bad quality figures. Bellow, there are the most obvious ones. 

Line 71 Please provide an explanation of what is a G1. It is the first appearance in a text.  

Figure 2 panel B) is xerocopy from the book SigmaProteins, edited by Felix J. Kim, Gavril W. Pasternak. As well as  Cell Tissue Kinet 1984, 17, 65-77. Please provide the reference. Also panel 2) has clear signs of xerocopy artifacts, please clean or request original image. 

Lines 64 through 93 have several direct copy-paste sentences from the aforementioned book. Please provide the correct references and rewrite the section.  

Line 95 There is a sharp change from the discussion  on 66P cell line to the Hela cell line.  Please provide a short discussion on why HeLa is a good candidate for TNEM97 protein targeting. 

Line 106  Please provide consistent metrics for the affinity.   Line 34 authors called affinity low at values >1000 nM and high at 3 nM. But in Line 106 affinity 302nM called relatively low. It is a very subjective, but will benefit to have consistent low, hight range agreement trough the text.   

Line 112 Figure 2 b, c are from Mach et al. 1997 not from Shoghi, K.I., et al. 2013

Figure 3 PET image of the uptake is cut.  The  SUVmax is usually presented as a unitless parameter. In the figure capture there is no explanation of what the left picture shows,  what right picture shows, and what bottom picture represents.  Please fix it. It is completely different from Figure 4 where all the panels are described and explained clearly.

Line 194  Localization of receptors is not a biological function by definition. It is a kinetic function. Although it is debatable. This section has to be relocated somewhere else.

Line 215 Expression of the receptor is not a biological function. This section has to be relocated somewhere else.

Reviewer 3 Report

The manuscript submitted by Zeng, Riad and Mach is a review article summarizing the biological function of the sigma-2-receptor and the utility in PET imaging studies of cancer. This review is rather broad and covers both results from cell studies, preclinical evaluation and clinical studies. Even so it is very broad; the review article is well written and is informative for an audience with limited knowledge in all aspects of the presented results.

As the referees main competence is on PET tracers and PET imaging, the main focus of the referee report will be on the PET part. Regarding PET references 17, 18 and 19 are the key references. Dynamic PET imaging and time activity curves presented in reference 17 indicate fast and irreversible uptake of F-18 ISO 1 in tumor. This uptake pattern justifies as well the reasoning of static scanning 60 min pots injection. This information should have been given in the review.

The preclinical results from reference 17 were confirmed in a patient study in reference 18. Here the fast and irreversible uptake was confirmed in dynamic scans. Furthermore reference 18 reports a high protein binding of F-18 ISO 1 and a slow metabolism. This information should have been given as well.

Figure 4 which was adopted from reference 21. Unfortunately reference 21 does not give reference to distinctive color scale. Here SUV would have been helpful. However, this is a weakness in reference 21, but the authors should consider the value of this image.

Regarding section 3.1.3. and the beginning of section 3.2.

The first section of 3.2. is based on the same reference 17 as in section 3.1.3. The authors should consider to move this section of section 3.2 to section 3.1.3.

Round 2

Reviewer 2 Report

na